# Exploring Community-Based Options for Reducing Youth Crime

**DOI:** 10.3390/ijerph18105097

**Published:** 2021-05-12

**Authors:** Kim Edmunds, Laura Wall, Scott Brown, Andrew Searles, Anthony P. Shakeshaft, Christopher M. Doran

**Affiliations:** 1Centre for Applied Health Economics, Griffith University, Nathan, QLD 4111, Australia; k.edmunds@griffith.edu.au; 2School of Psychology, University of Newcastle, Callaghan, NSW 2308, Australia; laura.wall@newcastle.edu.au (L.W.); scott.brown@newcastle.edu.au (S.B.); 3Health Research Economics, Hunter Medical Research Institute (HMRI), New Lambton Heights, NSW 2305, Australia; Andrew.Searles@hmri.org.au; 4National Drug and Alcohol Research Centre (NDARC), University of New South Wales, Randwick, NSW 2502, Australia; a.shakeshaft@unsw.edu.au; 5Cluster for Resilience and Wellbeing, Appleton Institute, Central Queensland University, Brisbane, QLD 4000, Australia

**Keywords:** youth crime, community, preference, intervention, economic

## Abstract

BackTrack is a multi-component, community-based intervention designed to build capacity amongst 14–17-year-old high risk young people. The aim of the current study seeks to explore community value and preferences for reducing youth crime and improving community safety using BackTrack in a rural setting in Armidale, New South Wales, Australia. The study design used discrete choice experiments (DCEs), designed in accordance with the 10-item checklist outlined by the International Society for Pharmacoeconomics and Outcomes Research. The DCE was pilot tested on 43 participants to test feasibility and comprehension. A revised version of the survey was subsequently completed by 282 people over a 12-day period between 30 May 2016 and 10 June 2016, representing a survey response rate of 35%. Ninety per cent of respondents were residents of Armidale, the local rural town where BackTrack was implemented. The DCE generated results that consistently demonstrated a preference for social programs to address youth crime and community safety in the Armidale area. Respondents chose BackTrack over Greater Police Presence 75% of the time with an annual benefit of Australian dollars (AUD) 150 per household, equivalent to a community benefit of AUD 2.04 million. This study estimates a strong community preference for BackTrack relative to more policing (a community willing to pay equivalent to AUD 2.04 million) highlighting the clear value of including community preferences when evaluating community-based programs for high-risk young people.

## 1. Introduction

The BackTrack program was established in Armidale in northern New South Wales (NSW), Australia, in 2006, for 14–17-year-old high risk young people (http://www.backtrack.org.au, accessed on 15 April 2021). It is underpinned by six key principles derived from previous reviews of the literature [1,2] together with feedback from staff [3]: (i) in recognition that its participants are more likely to engage in multiple risk behaviour, the program is comprised of multiple components that target different areas of need simultaneously (e.g., personal development, skills training and legal issues); (ii) flexibility in the delivery of the program components, which reflects that the focus of young people’s needs shifts over time; (iii) flexibility in program attendance, so that participants are able to start, leave, and re-enter the program as they wish, or as their life circumstances permit; (iv) a requirement that young people in the program eventually actively participate in all components of the program; (v) active engagement of local businesses, local media, key stakeholders (e.g., police, and magistrates), and community members in delivering program elements, resolving bureaucratic problems, providing infrastructure and funds, and facilitating communication about the benefits of the program; and (vi) recognition that achieving sustained change among high-risk young people will take a number of years.

The BackTrack program comprises a range of activities organised into five standardised core program components: effective engagement, to optimise participation in the program; individualised life-skills management, to address participants’ immediate and practical needs, such as attending court or finding secure housing; diversionary activities, to reduce participants’ exposure to high-risk situations, such as night-time encounters with police in public places or volatile situations at home; personal development, identity and team identity, to improve participants’ social and emotional (or psychological) wellbeing, increase their range of personal coping strategies (especially for using in high-risk situations) and to enhance their sense of connection to their peers and community; and learning and vocational skills, to increase their opportunities for active participation in education or training likely to lead to employment. This model of standardisation (the five core program components) with in-built flexibility (the specific activities that operationalise each component are selected and designed by staff), provides a mechanism to both standardise the intervention across multiple communities and tailor it to the resources available in different communities. Further details of the BackTrack program can be found on the BackTrack website (http://www.backtrack.org.au, accessed on 15 April 2021) and other sources [4].

Participants in the BackTrack program are typically at a high risk of drug and alcohol harm, psychological distress, and suicide. The eligibility criteria and the procedure for referral and acceptance into the program are detailed elsewhere [3]. Briefly, young people are eligible to participate in the service if they: (i) resided in a community where the service was available; (ii) were aged 14–21 years; and (iii) were currently experiencing more than one of the following behavioural risk factors: involvement in criminal activity; substance use; violent behaviour; homelessness; poor mental health and wellbeing; poor engagement with school (including suspensions and unexplained absences); and un- or under-employment. All program participants report experiencing risk factors in at least two domains of risk, and more than half experience risk factors in all four domains [3]. The most common risk factors were involvement in crime or with the juvenile justice system, school absence, unemployment, suicide ideation, psychological distress, substance use, low levels of physical activity, and low health service utilisation [3]. The presence of these risk factors places these young people at both short-term risk of harm and long-term risk of entrenched unemployment, criminal involvement, and incarceration [5,6,7,8,9]. Recent research has suggested that one in three serious young offenders strongly endorsed the view that crime had become their way of life with age of onset and frequency of offending reinforcing this view [10].

In addition to personal hardship, the harms experienced by these young people have a negative impact on their communities through increased social disruption, potential loss or damage to property, fear for personal safety, and increased health costs, as well as police, court and incarceration costs [11,12]. Given this impact on both individuals and the community, it is important to consider community value when evaluating the effectiveness or conducting an economic evaluation of a program like BackTrack. Such views have the potential to impact on program uptake, funding, sustainability and estimates of its benefits. Despite the importance of economic evaluation, a systematic review of community-based programs for high-risk young people found that no published evaluations to date have conducted an economic analysis or systematically quantified the costs or economic benefit derived from such programs [2]. The aim of the current study is to address this lack of data and seeks to explore community value and preferences for reducing youth crime and improving community safety using BackTrack in a rural setting in Armidale, New South Wales, Australia.

## 2. Materials and Methods

Discrete choice experiments (DCEs) provide a useful method for quantifying preferences for nonmarket goods and services [13,14]. DCEs are based on the premise that any good or service can be described according to its attributes, and the levels of these attributes determines the relative value they place on them. Respondents in a DCE must choose their preferred option from a group of hypothetical scenarios, called a choice set. It is proposed that aligning health care policy with patient or community preferences could improve the effectiveness of health care interventions by improving adoption of, satisfaction with, and adherence to clinical treatments or public health programs [15].

The application of DCEs has increased rapidly over the past decade and has been successfully applied to a diverse range of health applications including cancer treatment [16]; depression [17]; dermatology services [18]; diabetes [19] and treatments for Alzheimer’s disease [20]; and weight-loss programs [21]. To date, there has been limited application of DCE methods to community-based interventions for vulnerable youth such as the BackTrack program.

In order to develop consensus-based methodological standards, the International Society for Pharmacoeconomics and Outcomes Research (ISPOR) developed a 10-item checklist covering: research question; attributes and levels; construction of tasks; experimental design; preference elicitation; instrument design; data collection; statistical analyses; results and conclusions; and study presentation [22]. This study follows these guidelines. Although the checklist does not endorse any specific methodological approach to conjoint analysis such as choice of attributes or analytical approach, they do promote good research practices for the application of conjoint-analysis methods. For further discussion on methodological challenges inherent in conducting DCEs, the reader is referred to additional sources [23,24,25,26,27].

### 2.1. Research Question

The research question for the current study is: Does the community prefer community programs like BackTrack compared to greater police presence and what factors affect these preferences?

### 2.2. Attributes and Levels

Attributes and levels were identified using mixed methods. Researchers engaged in discussions with the research and BackTrack program teams in order to characterise the choice decision, identify the alternative programs, and determine which factors (attributes) might drive the decision process, and which were most important.

Given the tension between drawing a large sample from a regional area with a limited budget, the choice was restricted to three intervention options (BackTrack, greater police presence and current practice), each of which had two attributes (one for effectiveness and one for cost) and three levels of difference. This simple design helps to ensure adequate power to detect differences between the key variables of interest, program efficacy and cost. An additional literature search informed the identification of realistic values for these variables [1,2].

These measures led to a best-worst DCE design with three labelled options; current practice, BackTrack, and greater police presence, with two attributes each; less crime (as a percentage) and cost. The current practice option had fixed levels of 0% less crime and AUD 0 cost, while the BackTrack and greater police presence had three levels of 10%, 20%, and 30% and AUD 10, AUD 20, and AUD 40 for less crime and cost, respectively. These were initially arranged in a factorial design with each possible combination allowed, which was then reduced to a more efficient design. This design prohibited equal levels of one attribute across options, e.g., 20% less crime for both BackTrack and greater police presence, and removed dominating scenarios, e.g., where BackTrack was AUD 10 and 30% less crime while greater police presence was AUD 40 and 10% less crime. These dominating scenarios were defined by the research team.

This design was then tested in a pilot study of 43 participants to establish if the survey were easy to understand and complete, and to determine a magnitude of difference between the three levels of effectiveness that would allow respondents to easily discriminate and trade-off between percentage reductions in crime and cost. It was found that current practice was only chosen as the best option 2.7% of the time and it was chosen as the worst option 90.6% of the time. It was therefore removed for the final DCE as it was clearly a redundant option. Furthermore, it was found that people’s responses generally made sense in that higher costs and lower reductions in crime lead to a greater chance of selecting an option as best; however, the costs were too low to provide insight into the maximum value that people would be willing to pay. For the final DCE, the levels of cost were therefore adjusted to AUD 30, AUD 60, and AUD 120 based on model predictions from the pilot data. The pilot data did show differences between the efficacy levels however the difference between 20% and 30% tended to reduce as the cost increased; therefore, the third level of efficacy was increased slightly for the final DCE. Finally, there were some comments regarding the clarity and wording of the attributes which led to the descriptions seen in Table 1. Table 1 provides the components used to construct the choice sets for the final DCE.

### 2.3. Construction of Tasks

Each program type (BackTrack or greater police presence) generated nine possible scenarios (combinations of program characteristics and cost), so the possible pairs (i.e., choice sets) numbered 81. Choice sets with duplicate combinations of levels (n = 45) and with dominated scenarios (n = 18) were removed to minimise redundancy and maximise efficiency. Eighteen choice sets remained, the minimum number of choice sets required to incorporate every possible trade-off of the attributes and levels. These were organised into two blocks with identical structure, each containing half the choice sets (nine per block). The nine choice sets were then randomised within each block. To minimise respondent burden, participants were randomly allocated one of the two blocks.

### 2.4. Experimental Design

The removal of duplicate levels and dominating scenarios to increase the amount of useful preference information resulted in a non-factorial design. Across the two options of BackTrack and greater police presence there were six possible pairs of efficacy and six of cost, excluding duplicates, e.g., 10% for both options. A fully factorial design would then combine these six pairs with each other to produce 36 possible choice sets. Our design restricted the combinations of pairs such that an option could not have both a lower (or greater) cost and greater (or lower) percentage than the other option, and thus a trade-off needed to occur, as seen in Table 2. This resulted in each efficacy or cost pair occurring three times each across the 18 choice sets. Each combination of efficacy with cost, did not occur equally however with lower/higher costs more likely to occur with lower/higher efficacy. This can cause difficulties in interpreting marginal probabilities as it can appear that higher costs are preferred to lower costs (because they more often occur with higher efficacy); however, the cumulative link model used for analysis appropriately manages such a design.

### 2.5. Preference Elicitation

Preference elicitation was by trade-off. The respondent was asked to choose one option from the choice set, no allowance for indifference was provided. Explanations about how to complete the task and cheap talk were included. Background information also provided context of crime within the local catchment area. An example of the survey is provided in the Appendix A.

### 2.6. Instrument Design

The DCE utilised a survey for face-to-face delivery. Demographic and clinical background information was collected so that characteristics of respondents could be examined, and subgroup analyses could be performed, and responder bias examined. The survey included detailed descriptions on attributes and levels and an explanation of the decision scenario to provide good explanations in an attempt to inform participants, the choice they were asked to make, and how to complete the choice tasks.

### 2.7. Data Collection and Setting

Data was collected over a 12-day period between 30 May 2016 and 10 June 2016 by trained and experienced researchers. Data collection was scheduled at different times and days with researchers allocated to various locations to maximise reach. In order to minimise the potential for selection and volunteer bias, respondents were selected randomly by systematically counting passersby who were within the target age, to a pre-specified number and approaching the nth person (e.g., the 8th in a busy traffic area or 3rd in a low traffic area). The number of non-responders or individuals approached who did not consent to participate in the DCE was recorded to calculate the response rate. This record included gender and approximate age within a range, determined by the interviewer.

All researchers followed the same implementation protocol. They began with an introduction to the research project to elicit informed consent from potential respondents. In order to promote incentive compatibility, respondents were informed that the results of the study were part of an economic analysis that would be used to inform policy. Each respondent was then shown a sample choice set to ensure they understood the DCE survey format, the respective program attributes and efficacy and cost levels, as well as the task they were required to complete. Respondents were then asked to repeatedly choose between the two scenarios based on their efficacy and cost. After completing the nine choice sets, the respondents were given an additional, separate question which asked if there was an intervention or option other than BackTrack or greater police presence they would prefer. If the answer was affirmative, they were asked to explain their alternative. This question was followed by six demographic questions designed to elicit factors that may be influencing choice such as gender, age, employment status, education, and income. Postcode was included to confirm that respondents were residents of the specified community catchment.

### 2.8. Statistical Analyses

We used a cumulative/probit link model because it provides a psychological interpretation of the choices in terms of a Thurstonian, or random utility model, with normally distributed utility [28]. The DCE model excluded intervention-specific co-efficients for cost and benefit and excluded an alternative specific constant. The former specification was selected to facilitate ease of data collection and model interpretation, which was judged to be a higher priority than optimising the precision of the estimate, given this is the first economic modelling that has ever been applied to community-based programs for high-risk young people internationally [1,2]. The latter specification is appropriate given the aim of the study is to estimate preferences for existing approaches to reducing youth crime, rather than predicting the likely adoption of new strategies [23].

Statistical analysis of the data used a cumulative link model, an approach like logistic regression, to appropriately treat the non-factorial experimental design. We assumed a probit link, which also provides a psychological interpretation of the choices in terms of a Thurstonian, or random utility model, with normally distributed utility. We estimated a linear model on the mean of the utility distribution within the four factors of the design as predictors, variance fixed arbitrarily at 1, and an estimated decision threshold parameter (which measures overall bias towards or against BackTrack or Greater Police Presence). All analyses were conducted using R [29].

## 3. Results

A total of 282 respondents who satisfied the inclusion criteria completed useable DCE surveys. A total of 805 people were approached, a survey response rate of 35%. Women were more prevalent responders than men (60% vs. 40%), with slightly elevated numbers of female respondents in the 30–49-year age group. Ninety per cent of respondents were residents of Armidale (postcode 2350); of the remaining 10% in the survey catchment, 4% came from Uralla (postcode 2358) and 2.5% from Guyra (postcode 2365). The sample also comprised a variety of educational levels, with almost half (48%) having completed a university education, 42% of these with postgraduate qualifications. Only 14% of the sample had less than a Year 12 level of education. There was however, a more even spread of employment types and incomes. A total of 39% of respondents were in fulltime employment, 26% had part time or casual employment and 29% were not in the workforce. Of the 79% of the sample who answered the income question, 30% earned over AUD 80,000 per year and 27% earned AUD 39,000 or less.

Forty per cent of respondents responded affirmatively to the opt-out alternative question. Half of the suggestions provided by these respondents related to youth programs like BackTrack that addressed social needs, employment needs, educations needs or a combination of these. Eighteen per cent of respondents suggested family programs rather than youth programs; 10% opted for a combination of greater police presence and BackTrack; and 8% for mental health support or rehabilitation. The remainder suggested systemic, policy level changes; Indigenous specific; religion based; or justice-based approaches.

The DCE generated results that consistently demonstrated a preference for social programs to address youth crime and community safety in the Armidale area. Overall, the proportion of times in which respondents chose BackTrack over greater police presence was very high: 74.8%. The estimate of the decision threshold was significantly below zero, (−0.6385, z = −4.02), which reflects the strong tendency to favour BackTrack over greater police presence in these data. Each of the four attributes had a statistically significant effect on the utility, reflecting their importance to choose. The standardised regression coefficients (column “Z” in Table 3) indicates the relative importance of the different attributes. Two coefficients were positive (BackTrack benefit and police cost) indicating that greater values of those attributes led to more people choosing BackTrack over greater police presence, and two coefficients were negative (BackTrack cost and police benefit). These directions are exactly as expected. The effect on utility was greatest for BackTrack benefit, moderate for each of BackTrack cost and police benefit, and small for police cost. This pattern indicates that respondents were mostly influenced by the benefit of BackTrack. For example, adding one extra dollar to the cost of BackTrack has the same influence as subtracting approximately three dollars from the cost of police.

Estimated choice proportions from the probit regression are used to show how the different factors influence choice (Figure 1). Each of the four panels in the figure takes one of the factors and shows how the proportion of people choosing BackTrack changes as this factor is increased (while holding all the other factors constant at their median values). The dashed red lines show the point of indifference, 50% choice. The top left panel shows that people more often choose BackTrack than greater police presence (i.e., choice proportions above 50%) until the cost of the BackTrack option reaches approximately AUD 150. The top right panel shows that people always choose BackTrack more often than policing, no matter what cost we gave to the police option (at least for the median values of the other factors). The bottom left panel shows that BackTrack is mostly preferred over policing, even for very small benefits of the BackTrack option. The bottom right panel shows that respondents only choose policing more often than BackTrack once the benefit of the police option exceeds about 53%.

Analyses were re-run after splitting the data by gender (168 female, 112 male, and 1 unspecified), and by education level (96 with secondary education only, 185 with tertiary education, 2 unspecified), and by household income (110 with >AUD 60,000, 100 with <AUD 60,000, and 4 unspecified). For each subset, the probit regression above was recalculated. There were very small differences between the subsets. Men were more sensitive to cost than women, and this was true for the cost of the BackTrack program (men: z = −5.4; women: z = −3.5) and the cost of greater police presence (men: z = 2.4; women: z = 1.9). To make this concrete, women found BackTrack and police options equally attractive when the cost of BackTrack was AUD 196, but men found them equally attractive at a cost of only AUD 130 (using the median values of all other attributes, as above). People with tertiary education were more sensitive to all the factors than people with secondary education. However, this was not a reliable difference, most likely due to the relatively small number of respondents who had secondary education.

## 4. Discussion

In a rural/regional community where there is concern about youth unemployment and associated youth crime and antisocial behaviour [30], public perception of a community program designed to address the needs of high risk young people has the potential to be a powerful determinant of program acceptability, uptake, success, and sustainability. Our reviews of the literature identified a lack of outcome evaluation studies of interventions that targeted multiple risk factors, relative to single risk factors, among high-risk young people [1] or economic evaluations of interventions for high risk young people [2]. Further, very few studies have considered the viewpoint of the community itself or the value that they may attach to community-based programs that address youth crime.

This research leverages off the implementation of the BackTrack program implemented in Armidale, NSW, since 2006. Discrete choice experimental methodology was implemented to explore preferences for, and value of, implementing the BackTrack program to reduce youth crime and improve community safety.

### 4.1. Overall Findings

The results from this study showed that in a representative sample of the population of Armidale, there was strong preference for BackTrack. Overall, respondents chose BackTrack over greater police presence 75% of the time and continued to choose BackTrack to a cost of AUD 150 per household per year equivalent to a total benefit of AUD 2.04 million per annum.

The effectiveness of BackTrack was the strongest predictor of choice; however, all four attributes or predictors (the cost and benefit for each of the two choice options—BackTrack and greater police presence) had a statistically significant effect on utility. The direction of the effect of the four attributes on utility was as expected. For example, greater BackTrack benefit and greater police cost resulted in respondents choosing BackTrack over greater police presence, whereas greater BackTrack cost and greater police benefit had a negative effect on the choice of BackTrack over greater police presence. Split sample analyses, conducted to account for differences in preferences that arise from differences in individual characteristics such income, education, and gender, revealed no significant differences between the subsets analysed. These results provide further evidence of the strong community support for BackTrack. Interestingly, when respondents were given the option of suggesting an alternative program to the two offered in the DCE, most suggestions (approx. 80%) were similar social/education/health programs or a combination of greater police presence and BackTrack.

This research fills a void in the literature in terms of understanding community values and preferences for programs like BackTrack. Such information is important in the context of program uptake, funding, and sustainability. The research methods also extend the application of DCE methods and provide important inputs into economic evaluations by valuing community benefit. The Armidale community has embraced the BackTrack program and graduates of the BackTrack program are seen as important community members [4]. This is in stark contrast to community attitudes of participants first enrolling in the program that have a legacy of crime and community disruption.

### 4.2. Limitations

Data collection was somewhat limited by time and funding constraints; however, a sample size of 282 was considered adequate for the purposes of the analysis. This study removed the opt-out alternative because it was judged to be the worst alternative by 90% of respondents. Excluding choices/pairs in a labelled DCE needs great care because of relatively subjective judgements about the threshold for what proportion of responses ought to be properly regarded as a marginal result, and because retaining more alternatives can be used to help more precisely understand the outcomes. In this study, for example, we have estimated respondents’ willingness to pay for different interventions but retaining more alternatives could have also helped determine whether respondents made their choices because of their preferences related to the effectiveness of each alternative or the type of intervention (e.g., some respondents may be willing to pay more for greater police presence even though it is less effective because they simply prefer more police on the streets). Future research could start to examine the decision-making process of respondents’ in determining their preferences in addition to identifying their preferences per se.

We used a cumulative/probit link model because it provides a psychological interpretation of the choices in terms of a Thurstonian, or random utility model, with normally distributed utility [28]. Although the cumulative/probit model was selected because of its choice and technical features were appropriate, a mixed model could have been used to demonstrate the distribution of preferences across the population. Future research could utilise both approaches to quantitatively examine the robustness of the results to the choice of model and analysis.

The key design features of this study were the exclusion of intervention-specific co-efficients for cost and benefit, and the exclusion of an alternative specific constant. Having a cost-specific coefficient (separately to a co-efficient for effectiveness) would allow a more precise estimation of WTP and should be integrated into future DCE evaluations of programs for high-risk young people. Although the inclusion of an alternative specific constant is generally recommended in DCEs to avoid forced choices, the decision to exclude it in this study was appropriate for two reasons. First, neither of the alternatives under consideration were new services in the community, meaning that respondents were asked to choose between realistic, existing alternatives rather than more abstract experimental options for which an opt-out option would be appropriate. Second, the way in which an opt-out alternative could have been presented to respondents would have been arbitrary and of unknown impact on respondent’s choices [23].

A further methodological issue is that the extent to which the results can be reasonably extrapolated to the whole community might be limited by two factors. First, this DCE only considered one alternative (BackTrack) to more policing, rather than multiple alternative options, such as harsher penalties for offending or increasing youth detention. The extent to which the community would preference BackTrack over alternatives other than more policing across the entire population of the community remains unclear. Second, is the extent to which the sample was representative of the population. A response rate of 35% was lower than expected but, given the unfamiliar nature of a DCE survey and the potential for respondent burden, was an acceptable outcome. The survey results showed that the DCE captured a representative sample of the population of the Armidale region [31]. The largest proportion of non-responders were women in the 30–49-year age group; a group who tended to be apologetic, citing lack of time, being at work, busy with children or going to collect children, as their reasons for not participating.

## 5. Conclusions

This study estimates a strong community preference for BackTrack relative to more policing (a community WTP of AUD 2.04 million). Although it is a compelling result, the exact strength of the estimated preference may lack some precision as a consequence of the methods. Nevertheless, the apparent strong preference for community-based programs to reduce youth crime relative to more policing, coupled with the new availability of refined DCE methods [23,27], highlights the clear value of replicating this DCE with more community-based programs for high-risk young people.

Although the BackTrack program commenced in Armidale, the program has been implemented in several other rural communities. The BackTrack strategy is to build capacity and capability to positively impact the lives of many more young people across a range of disparate communities. Building an evidence base for programs like BackTrack are essential for ongoing investment and sustainability. This research adds to this evidence base by highlighting strong community preferences for youth based programs that are community based rather than traditional means of dealing with youth crime through punitive measures.

## Figures and Tables

**Figure 1 ijerph-18-05097-f001:**
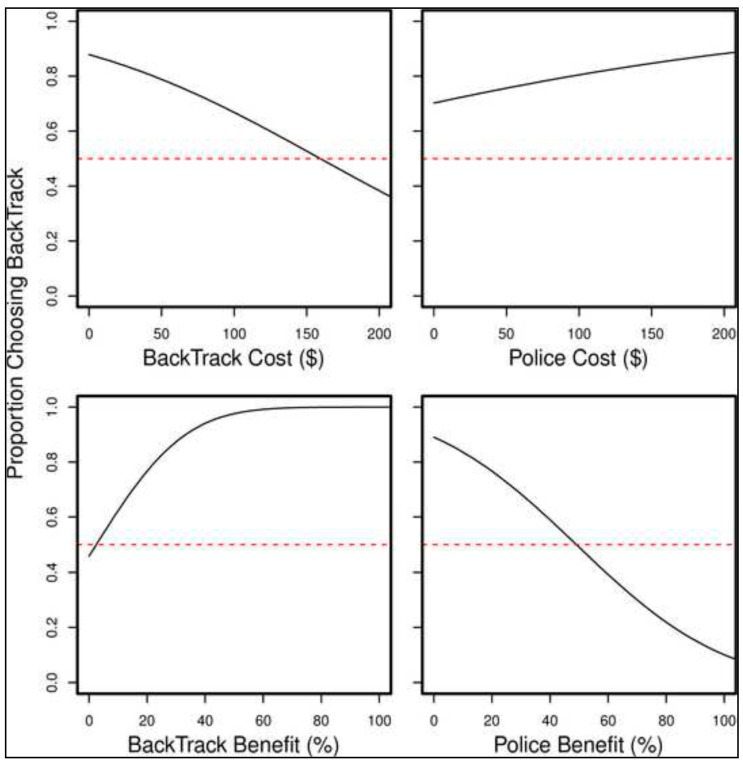
Proportion of sample choosing BackTrack.

**Table 1 ijerph-18-05097-t001:** DCE choices, attributes, and levels.

Program/Intervention Choice	Attributes	Levels for Attributes
BackTrack	1. Reductions in crime and improvements in health and educational outcomes	1. 10%
		2. 20%
		3. 35%
	2. Annual cost to household	1. AUD 30
		2. AUD 60
		3. AUD 120
Greater police presence	3. Reductions in crime and improvements in health and educational outcomes	1. 10%
		2. 20%
		3. 35%
	4. Annual cost to household	1. AUD 30
		2. AUD 60
		3. AUD 120

**Table 2 ijerph-18-05097-t002:** Example choice sets that were included and excluded from the design due to dominating options.

Type	BackTrack	Greater Police Presence	Dominating?
Efficacy	20%	10%	BackTrack dominates (not included)
Cost	AUD 30	AUD 60	
Efficacy	10%	20%	Police presence dominates (not included)
Cost	AUD 60	AUD 30	
Efficacy	10%	20%	Neither dominates (included)
Cost	AUD 30	AUD 60	
Efficacy	10%	35%	Neither dominates (included)
Cost	AUD 30	AUD 120	

**Table 3 ijerph-18-05097-t003:** Statistical Analysis.

	Coefficient	St. Error	Z	*p*
BackTrack cost	−0.007323	0.001202	−6.094	<0.0001
Police cost	0.003277	0.001095	2.993	0.00276
BackTrack benefit	0.041621	0.004306	9.666	<0.0001
Police benefit	−0.025035	0.004145	−6.04	<0.0001

## Data Availability

No restrictions apply to the availability of these data and are available from the corresponding author.

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
