# Peer review of "Exploring Community-Based Options for Reducing Youth Crime"

_ijerph, 2021, doi:10.3390/ijerph18105097_

Round 1
Reviewer 1 Report
The reviewed article is well structured. Nevertheless, it should:
1.The introduction be expanded and the literature be enriched. For
example, in line 63, where there is mention of "a negative impact" on the
communities.
2.The discussion is treated in a similar way. For the reviewer it is too
general, which makes it contribute little to the article as a whole.
3. The dates of the research are unclear. Neither lines 185 to 188 help to define them, nor the very vague information given in lines 205 to 206. The reviewer insists that the time when the research was carried out and the justification for its validity should be made clearer.
Author Response
Please see atachment

Reviewer 2 Report
Dear author(s),
Your paper was an interesting reading. I really enjoyed the DCE method, which I was not aware of before reading your article. I also could not find any mistakes in your writing. In fact, I quite enjoyed my reading.
I have only some suggestions to your manuscript. Generally, I recommend you add more detail in describing the BackTrack and in describing your methodology. In fact, the major problem of the article is that even now, after reading the article, I still do not have a clear idea of what the BackTrack program actually is. In fact, I had to look it up online (https://backtrack.org.au/). That is very serious issue given the centrality of the program for the study, but I also believe this is a simple limitation to address by adding a paragraph in beginning of the manuscript describing the program.
You succeeded in showing evidence of individuals’ preference to different strategies of addressing crime, which is an interesting contribution. I will use the same method in my own research.
- Describe what the program actually is and does, before describing its objectives in the first paragraph. I finished reading the introduction without a clear definition of the program. You should clearly make that description as soon as possible in your manuscript, perhaps in the very first paragraph.
- (2nd paragraph) How are the participants assessed? I suggest a single sentence.
- Because I did not have prior knowledge of the methodology (Discrete Choice Experiments), the reading of the Materials and Methods section was difficult and somewhat unpleasant. I only understood the method when I saw the authors actually implementing it. I suggest one more paragraph clearly describing the premises behind the DCE method, the selection of options, and its nuances. I recommend that the authors add this description in the beginning of the Material and Methods section
Author Response
Pleas see attachment

Reviewer 3 Report
The paper is very interesting for readers. It is easy to understand what authors wanted to present. Results are presented in very clearly. For me two elements should be improved:
- the aim of the paper - it is not written very clearly,
- conclusion - for me it is too short.
Reviewer 4 Report
I recommend that the BackTrack community intervention be better explainedAuthor Response
Please see attachment
